# Hepatitis E Virus Infections among Patients with Acute Febrile Jaundice in Burkina Faso

**DOI:** 10.3390/v11060554

**Published:** 2019-06-14

**Authors:** Chloé Dimeglio, Dramane Kania, Judith Mbombi Mantono, Thérèse Kagoné, Sylvie Zida, Souleymane Tassembedo, Amadou Dicko, Bachirou Tinto, Seydou Yaro, Hervé Hien, Jérémi Rouamba, Brice Bicaba, Isaïe Medah, Nicolas Meda, Oumar Traoré, Edouard Tuaillon, Florence Abravanel, Jacques Izopet

**Affiliations:** 1Laboratoire de virologie, Centre national de référence du virus de l’hépatite E, CHU Toulouse, Hôpital Purpan, 31300 Toulouse, France; abravanel.f@chu-toulouse.fr (F.A.); izopet.j@chu-toulouse.fr (J.I.); 2Centre MURAZ, Bobo-Dioulasso, Burkina Faso; draka3703@yahoo.fr (D.K.); mantonochancelvie@yahoo.fr (J.M.M.); tskagone@gmail.com (T.K.); zida_sylvie@yahoo.fr (S.Z.); tassoulee@yahoo.fr (S.T.); dicko22@gmail.com (A.D.); tintobachirou@yahoo.fr (B.T.); yaro_seydou@yahoo.com (S.Y.); hien_herve@hotmail.com (H.H.); jeremirouamba@gmail.com (J.R.); 3Université Catholique d’Afrique de l’Ouest, Bobo-Dioulasso, Burkina Faso; 4Institut de Recherche en Sciences de la Santé (IRSS), Ouagadougou, Burkina Faso; bicaba_brico@yahoo.fr; 5Université Ouaga I Pr Joseph KI-ZERBO, Ouagadougou, Burkina Faso; 6Ministère de la Santé, Ouagadougou, Burkina Faso; isaiemedah@yahoo.fr (I.M.); nicolas.meda@gmail.com (N.M.); 7Agence nationale de biosécurité, Ouagadougou, Burkina Faso; kourouda@gmail.com; 8Pathogenesis and Control of Chronic Infections., Etablissement Français du Sang, CHU Montpellier, INSERM, University of Montpellier, 34090 Montpellier, France; e-tuaillon@chu-montpellier.fr; 9Centre de Physiopathologie de Toulouse Purpan (CPTP), UMR Inserm, U1043, UMR CNRS, U5282, 31300 Toulouse, France

**Keywords:** Burkina Faso, hepatitis E virus, epidemiology, risk factors

## Abstract

Hepatitis E virus infection is a significant public health problem in many parts of the world including Africa. We tested serum samples from 900 patients in Burkina Faso presenting with febrile icterus. They all tested negative for yellow fever, but those from 23/900 (2.6%) patients contained markers of acute HEV infection (anti-HEV IgM and HEV RNA positive). Genotyping indicated that 14 of the strains were HEV genotype 2b. There was an overall HEV IgG seroprevalence of 18.2% (164/900). In a bivariate analysis, the factors linked to HEV exposure were climate and patient age. Older patients and those living in arid regions were more likely to have HEV infection. HEV genotype 2b circulating only in humans can be involved in some acute febrile icterus cases in Burkina Faso. Better access to safe water, sanitation, and improved personal hygiene should improve control of HEV infection in this country.

## 1. Introduction

Hepatitis E virus (HEV) is a leading cause of viral hepatitis worldwide; an estimated 20 million people are infected every year, leading to 3.3 million symptomatic cases of hepatitis E [1]. HEV is a member of the Hepeviridae family, genus Orthohepevirus, whose species (A–D) infect terrestrial vertebrates but have very different host ranges [2,3]. Humans are infected by four major genotypes of Orthohepevirus A. HEV genotypes 1 and 2 infect only humans and are prevalent in low-income countries. The virus is transmitted mainly via the faecal-oral route; poor sanitation allows the virus in the faeces of infected people to reach drinking water supplies. The HEV genotypes 3 and 4 typically cause sporadic cases in developed countries. Their transmission is zoonotic. A large number of animals have been found to carry HEV, but pigs are the main reservoir of the virus [2,3]. Of note, genotype 7 was found in camels. It has caused a chronic infection in an immunocompromised patient who regularly ate camel products [4]. Hepatitis E virus infections are a significant public health problem in Africa where they can cause large outbreaks of acute hepatitis in displaced-person camps [5,6,7,8,9,10,11,12,13,14,15]. While the disease is usually self-limiting, HEV genotype 1 can kill up to 20% of infected pregnant women [13,16]. 

Understanding HEV epidemiology in Africa will facilitate the implementation of evidence-based control policies designed to prevent the spread of HEV. Burkina Faso, which lies between the Sahara desert and the Gulf of Guinea, is green in the south, with forests and fruit trees, and desert in the north. This low-income country has three climatic zones: the Sahel arid zone, the Sudan-Sahel semi-arid zone, and the Sudan-Guinea wetlands. A first survey of the zoonotic risk of Hepatitis E virus (HEV) transmission in Ouagadougou, Burkina Faso was conducted. They found that anti-HEV antibodies were more prevalent in pork butchers, than in the general population. Among slaughter-aged swine, HEV seroprevalence was of 80%, and HEV RNA was detected in 1% of pork livers. Phylogenetic analysis pointed out HEV genotype 3 [17], but little is known about the epidemiology and the genotype causing symptomatic hepatitis E in humans in this country.

We have investigated the presence of HEV in the cases of acute febrile jaundice reported in the Burkina Faso Yellow Fever (YF) surveillance and identified the involvement of the HEV genotype.

## 2. Materials and Methods

We tested 900 serum samples collected between 2013 and 2016 from patients identified by the Burkina Faso Yellow Fever (YF) national surveillance presenting with a clinical picture of the febrile icterus. All the samples were tested negative for YF and were tested retrospectively for HEV if the volume was sufficient. We analysed notification forms (which included the date of jaundice onset, age, gender, and residential district). The patient’s sex and age, domicile (urban or rural), climate zone (arid, semi-arid, or tropical savannah), as well as the year of sampling were analysed for their potential association with markers of an acute HEV infection (HEV RNA and/or anti-HEV IgM positive) and exposure to HEV (HEV RNA and/or anti-HEV IgM and/or anti-HEV IgG positive). Blood human samples were taken after obtaining the informed consent for the YF surveillance. Biological material and clinical data were obtained only for standard viral surveillance (no specific sampling, no modification of the sampling protocol, no questions in addition to the standardized questionnaire). Data were analysed using an anonymized database. According to the Public Health French law (CSP Art L 1121-1.1), such a protocol does not require written informed consent. 

All samples were tested for anti-HEV antibodies and HEV RNA. Wantai HEV IgG and IgM EIA kits (Wantai Biologic Pharmacy Enterprise, Beijing, China) were used as set out in the manufacturer’s instructions. Results were considered to be negative if the IgG or IgM index was <1 and positive if the index was ≥1. HEV RNA was detected using the Procleix HEV RNA assay (Grifols, Barcelona Spain) implemented on the Panther automate (limit of detection: 7 IU/mL) [18,19]. HEV RNA was quantified using an accredited ISO15189 HEV RNA assay (lower limit of detection: 53 IU/mL) [18,20]. The HEV genotype was determined by sequencing a fragment of the ORF2 genome [21] and by phylogenetic analysis based on the reference sequences proposed by Smith et al. using the ClustalX software [22].

Anonymised questionnaire data were entered into an Excel file and analysed using the Stata version 14 (StataCorp LP, College Station, TX, USA). Demographic factors associated with exposure to HEV were evaluated using bivariate analyses. Chi-squared and Fisher’s exact tests (when the expected cell frequency was less than 5) were used for binary variables. Student’s *t* test was used for normally distributed quantitative variables. Statistical significance was set at *p* < 0.05.

## 3. Results

The cohort consisted of 526 males (58.4%), the mean age was 15 years (±0.5, 95% CI: 14–16.1). Of the 900 patient samples tested, we found 23/900 (2.6%; 95% CI: 1.6%–3.8%) who had markers of an acute HEV infection. The 23 samples included 15 that were anti-HEV IgM and HEV RNA positive, four that were anti-HEV IgM negative and HEV RNA positive, and four that were anti-HEV IgM positive and HEV RNA negative (Figure 1). 

Median HEV RNA concentrations was 4.8 log10 UI/mL and ranged from 1.8 to 6.5 log IU/mL. The 14/19HEV RNA positive samples that could be genotyped were all genotype 2 (Genbank accession number MK412900-MK412913) (Figure 2) and 86.4–94.8% homologous with the HEV subtype 2b recently identified in Nigeria (Genbank accession number MH809516) (Figure 2). 

In a bivariate analysis, age was the only variable linked to a recent HEV infection (unadjusted OR = 1.02%–95% CI [1–1.05], *p* < 0.01). Patients with symptomatic acute hepatitis E were older than the other patients and more likely to be male: 18 of 23 (78.3%) versus 508 of 877 (57.9%) (*p* = 0.06) (Table 1). Tests for anti-HEV IgG in the 900 samples from patients presenting with fever and icterus indicated that 164 had been exposed to HEV (seroprevalence: 18.2%; 95% CI: 16.3%–21.5%) (Figure 1). Patient age and climate were associated with HEV exposure (*n* = 169, Figure 1) (Table 2). The prevalence of anti-HEV antibodies increased with patient age (Figure 3). It was also higher in patients living in the arid, mainly Northern, area (26.3%) than in the semi-arid and tropical areas (11.3%; OR = 2.8 (95% CI: 1.58–4.97) (Table 2). 

## 4. Discussion

Our data indicate that HEV was implicated in 2.6% of the cases of acute febrile jaundice identified in Burkina Faso during 2013–2016. HEV genotype 2b was detected in samples that could be genotyped and 18.2% of this population had anti-HEV IgG.

But relatively few of the patients with acute febrile jaundice in Burkina Faso had suffered from acute hepatitis E. Of note, the contribution of other pathogens such as hepatitis B or C virus was not investigated in the present population. A recent study in the Democratic Republic of Congo found that 10.4% of 365 patients with febrile jaundice had markers of HEV infection [23], but this could be due to the increased frequency of HEV infection in a single town in August 2006 caused by an unnoticed hepatitis E outbreak [23]. Unfortunately, the HEV genotype was not investigated in this latter study. Many of the acute HEV infections that occur in Africa are the result of outbreaks among displaced people [13,15]. 

A previous study in Burkina Faso found HEV genotype 3 in swine [17]. It also found that many (76%) of the pork butchers from the Ouagadougou area tested positive for anti-HEV IgG, indicating that transmission was zoonotic in this country [17]. All the HEV-infected patients in the present study had encountered the human-specific genotype 2b, suggesting that genotype 2b contributes to most of the symptomatic acute HEV infections with febrile jaundice. This also suggests that there are several sources of contamination in Burkina Faso. Transmission can be zoonotic with pigs as a reservoir of HEV genotype 3, as in high-income countries. But HEV genotype 2 transmission can be waterborne, due to using water contaminated with faeces [24,25]. HEV genotype 2 has been identified in several African countries: in an outbreak in the Central African Republic [26], in Namibia [14], in a refugees camp in Chad [6], and also in recent outbreaks in Nigeria [9]. Thus this subtype of HEV, which is different from the subtype 2a identified in Mexico, is firmly established throughout the continent. HEV genotype 1 also circulates in several African countries: Central African Republic [26], Chad [26], Sudan [27], Egypt [28,29], Uganda [12], and Namibia [30]. But, the zoonotic HEV genotype 3 was also observed in South Africa [31] Nigeria [32] Egypte [15], and Madagascar [33] suggesting that both waterborne and zoonotic transmissions occur in Africa.

The mean age of patients with markers of an acute symptomatic HEV infection was 22 years, while that of uninfected patients was 15 years (*p* = 0.03). Furthermore, exposure to HEV increased regularly with age, to reach 50% in those over 40 years old. This agrees well with studies in low-income Asian countries where genotype 1 is prevalent. HEV-1 is considered to be an infection of young adults, with a peak incidence in subjects aged between 15 and 35 years [3,13]. A higher number of symptomatic infections in young adults versus children has always been reported for genotype 1 and 2 in developing countries [13]. The mechanism explaining this difference of symptomatic cases according to age and HEV genotype is not known and deserves further studies.

The prevalence of anti-HEV IgG among our 900 Burkina Faso patients was 18.2% (95% CI: 16.3%–21.5%), similar to the 19.1% prevalence of anti-HEV IgG among 89 blood donors and 11.6% prevalence among 189 pregnant women found by Traoré et al. in 2012 using the Dia-pro anti-HEV assay [34]. A 2014 study using the Dia-pro assay found that 39% of 1497 blood donors were HEV IgG-positive [35], more than twice of that found in the present study. But like us, the authors found no significant difference between the seroprevalence in male and female donors. This prevalence of anti-HEV IgG could be linked to zoonotic transmission with HEV genotype 3 and waterborne transmission with HEV genotype 2 in Burkina Faso. 

Our rural and urban subjects had similar frequencies of anti-HEV markers, despite the findings in other African countries. One study found that HEV was about 2.0 times more prevalent in urban areas of Gabon (13.5%) than in rural areas (6.4%) [36]. In contrast, HEV markers were more prevalent (15.3%) among the rural residents of South Africa than among urban ones (6.6%) [37]. Our data indicate that anti-HEV IgG was more prevalent among patients living in the arid north of Burkina Faso, both rural and urban. This may be due to a combination of limited water supply and poor sanitation. 

We conclude that the genotype 2b infections identified in our patients indicate that the source of these sporadic cases of HEV is different from the HEV genotype found previously in pigs in Burkina Faso. If so, controlling this disease will also require better access to safe water, sanitation, and improved personal hygiene.

## Figures and Tables

**Figure 1 viruses-11-00554-f001:**
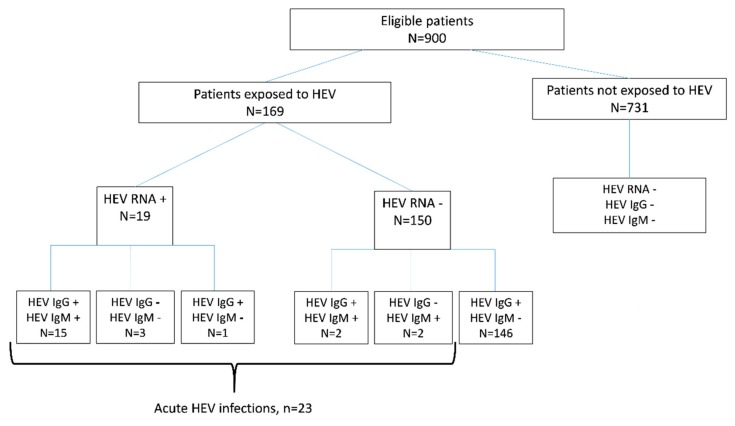
Study flow chart.

**Figure 2 viruses-11-00554-f002:**
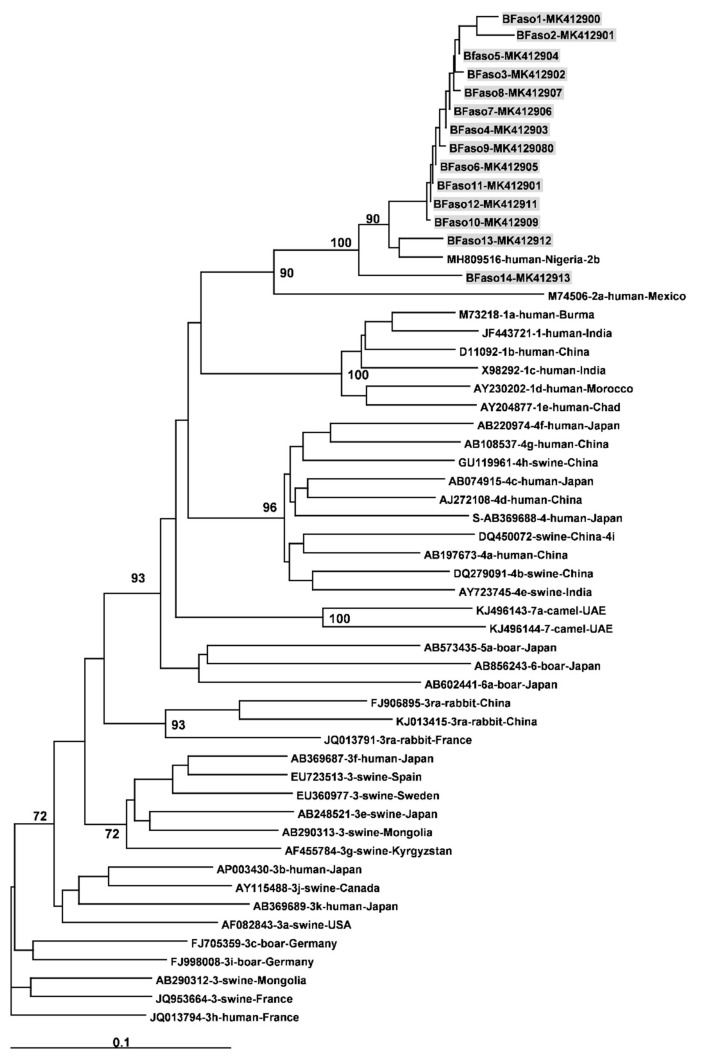
Phylogenetic tree constructed using 347-nt-long partial sequences within ORF2 (black dots). Genetic distances were calculated using the Kimura two-parameter method, phylogenetic trees were plotted by the neighbour-joining method. Bootstrap values acquired after 100 replications are shown (branch lengths measured as the number of substitutions per site). Highlighted patient sequences (grey boxes) were compared to reference sequences (Smith et al., 2016) and to the new full length of subtype 2b strain (MH809516). Accession numbers, genotypes, and country of origin of collections are listed.

**Figure 3 viruses-11-00554-f003:**
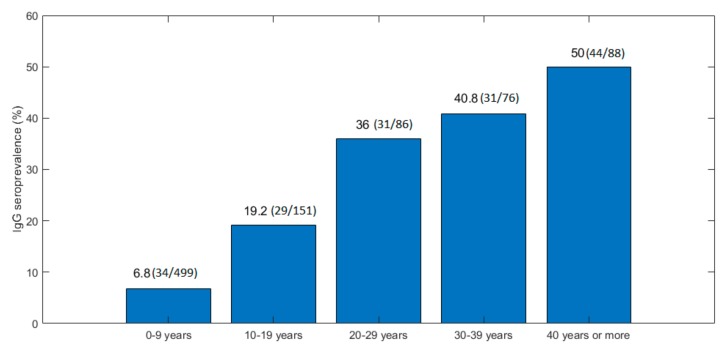
Prevalence of anti-HEV IgG in patients by age group.

**Table 1 viruses-11-00554-t001:** Demographic characteristics of patients with markers of a recent HEV infection (anti-HEV IgM and/or HEV RNA positive).

Variable	Evidence of Recent Infection (*n* = 23)	No Evidence of Recent Infection (*n* = 877)	*p*-Value
**Age, years**			0.03
Mean (standard error)	22 (2)	15 (0.6)	
95% CI	[18.3–26.3]	[13.8–16]	
Sex			0.06
Male	18 (3.4%)	508 (96.6%)	
Female	5 (1.3%)	369 (98.7%)	
**Year of sampling**			0.08
2013	0 (0%)	106 (100%)	
2014	12 (4%)	288 (96%)	
2015	6 (3.1%)	188 (96.9%)	
2016	5 (1.7%)	295 (98.3%)	
**Living area**			0.60
Urban	20 (2.3%)	706 (97.7%)	
Rural	3 (1.7%)	171 (98.3%)	
**Climate/Region**			0.07
Arid/North	5 (3.7%)	129 (96.3%)	
Semi-arid/Centre	9 (4.2%)	206 (95.8%)	
Tropical/South	9 (1.6%)	542 (98.4%)	

**Table 2 viruses-11-00554-t002:** Demographic characteristics of patients according to HEV exposure.

Status:	Patients Exposed to HEV (*n* = 169)	Patients Not Exposed to HEV (*n* = 731)	*p*-Value
**Age**			<0.01
Mean (standard error)95% CI	30(1.5)[27.9–33.4]	12(0.5)[10.5–12.5]	
Sex			0.36
Male	104 (19.8%)	422 (80.2%)	
Female	65 (17.4%)	309 (82.6%)	
**Year of sampling**			0.28
2013	25 (22.9%)	81 (77.1%)	
2014	62 (20.2%)	238 (79.8%)	
2015	32 (16.5%)	162 (83.5%)	
2016	50 (16.6%)	250 (83.4%)	
**Living area**			0.94
Urban	136 (18.7%)	590 (81.3%)	
Rural	33 (19%)	141 (81%)	
**Climate/Region**			<0.01
Arid/North	35 (26.3%)	99 (73.7%)	
Semi-arid/Centre	110 (19.9%)	442 (80.1%)	
Tropical/South	24 (11.3%)	190 (88.7%)

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
