# Peer review of "Hepatitis E Virus Infections among Patients with Acute Febrile Jaundice in Burkina Faso"

_viruses, 2019, doi:10.3390/v11060554_

Reviewer 1 Report

Dr. Dimeglio and colleagues presented an interesting study on the analysis of HEV as contributor to cases of febrile jaundice in Burkina Faso. The colleagues analysed 900 yellow fever negative samples collected by the Burkina Faso Yellow Fever national surveillance from patients presenting with febrile jaundice between 2013 and 2016 for the presence of markers of acute HEV infection or general exposure to HEV by means of ELISA and PCR. Moreover, the authors characterised the genotype of samples tested HEV RNA positive.

As a result the authors detected an overall HEV IgG seroprevalence of 18.2% while 2.6% of the analysed samples contained markers of an acute HEV infection. Within these, HEV subtype 2b was identified exclusively, leading to the conclusion that besides zoonotic transmission due to HEV-3 which was suggested previously by others researchers, also transmission via the faecal-oral route is of importance in Burkina Faso.

The study is well performed, written concisely and is a good addition to the knowledge on circulating HEV genotypes on the African continent. However, there are some points which should be addressed.

Specific comments

1. Introduction, line 38: The colleagues write about 5 HEV genotypes infecting humans. However, genotype 7 so far only caused infection in one single case. Therefore, it might be more appropriate to describe the 4 human-infecting HEV genotypes (1-4) and to refer to this single case additionally.

2. Material and Methods, line 60: Missing word: “All the samples were tested negative for YF and were tested retrospectively for HEV”.

3. Material and Methods: Are the 900 analysed samples the total amount of yellow fever negative samples collected between 2013 and 2016 or only a selection and if so, which criteria were used for selection?

4. Material and Methods, line 62: The colleagues state that “geographical region and climate zone” were analysed. However, in the results section including tables 1 and 2 no reference to different geographical regions has been described. Therefore, this point should be presented in more detail and included in the tables.

5. Material and Methods, line 62-65: The sentence is rather difficult to understand and it is not clear which further (?) epidemiological factors are meant. Please clarify. In line 63-64 it should read “(HEV RNA and/or anti-HEV IgM positive)”.

As a suggestion, a better wording could be: “The patient’s sex and age, domicile (urban or rural), climate zone (arid, semi-arid or tropical savannah) as well as the year of sampling were analysed for their potential association with markers of an acute HEV infection (HEV RNA and/or anti-HEV IgM positive) and exposure to HEV (HEV RNA and/or anti-HEV IgM and/or anti-HEV IgG positive).”

6. Results, line 80-81: The insertion “526 males (58.4%), mean age 15 years (±0.5, 95% CI: 14-16.1)” interrupts the reading flow. It would be better to characterise the analysed cohort in a first sentence and highlight the results in the following sentence.

7. Results, Figure 1: It would be helpful to highlight the boxes including the 23 acute infections by colour or bold outlines.

8. Results, line 88: “14/23” indicates that all 23 samples could have been genotyped. However, 4 of these samples lacked HEV RNA. Therefore, “14/19 HEV RNA positive samples” might be more appropriate.

9. Results, Figure 2: Are the sequences submitted to GenBank? If not, please, do this.

10. Results, Figure 2: The software which was used for generating the phylogenetic tree should be indicated.

11. Results, Figure 2: Additional African HEV-2b sequences (to which the colleagues also refer to in the discussion section in line 131) should be added to the phylogenetic analyses to better resolve the phylogenetic relationship of this subtype. And, please, indicate the sample numbers or ID of the patients at the dots of the phylogenetic tree.

12. Results, Figure 3: The number of samples and the number of IgG-positive cases for each group should be indicated.

13. Discussion, line 114: The statement “Our data indicate that HEV genotype 2 was implicated in 2.6% of the cases of acute febrile jaundice” is not correct. Of the 23 cases only 14 could be genotyped.

14. Discussion, line 125-127: As the samples were not tested for HBV and HCV as well as other possible pathogens the febrile icterus may not necessarily be attributable to HEV although markers of acute HEV were identified. This should be stated as a limitation of the study.

15. Discussion, line 129-133: The colleagues describe HEV-2 being identified in a number of African countries (line 130) and conclude that subtype 2b is established throughout the continent (line 133). It should be indicated more clearly, that the HEV-2 sequences to which they are referring are of subtype 2b which could be very nicely supported by including those sequences in the tree.

16. Discussion, line 137: The mean age of 14 is not equal to the one stated in line 80 (mean age=15).

17. Discussion, line 153-154: There is no hint to conclude that the source of the HEV-2b infections found in this study is different from previous cases in Burkina Faso, as no previous cases are reported.

The high seroprevalence found in butchers and the detection of HEV-3 in pigs in an earlier study indicates that probably asymptomatic or subclinical infections due to zoonotic transmission might be common in Burkina Faso.

However, the finding in this study indicates that acute, symptomatic HEV infections are rather due to faecal-oral transmission. Due to the lack of HEV surveillance in Burkina Faso the finding and interpretation of symptomatic cases is incomplete since no other study analysed HEV genotypes in Burkina Faso so far. Therefore, the conclusion should better highlight the confirmed circulation of subtype 2b in Burkina Faso.

Author Response

1.      Introduction, line 38: The colleagues write about 5 HEV genotypes infecting humans. However, genotype 7 so far only caused infection in one single case. Therefore, it might be more appropriate to describe the 4 human-infecting HEV genotypes (1-4) and to refer to this single case additionally.

We have modified the introduction as suggested by the reviewer line 38.

2.      Material and Methods, line 60: Missing word: “All the samples were tested negative for YF and were tested retrospectively for HEV”.

We have completed the sentence as suggested.

3.      Material and Methods: Are the 900 analysed samples the total amount of yellow fever negative samples collected between 2013 and 2016 or only a selection and if so, which criteria were used for selection?

There was no selection. All the yellow fever negative samples collected between 2013 and 2016 were tested if the volume was available.

4.      Material and Methods, line 62: The colleagues state that “geographical region and climate zone” were analysed. However, in the results section including tables 1 and 2 no reference to different geographical regions has been described. Therefore, this point should be presented in more detail and included in the tables.

Indeed, the variable « geographical regions » was included among the variables collected in our study. However, it was not analysed as it could be redundant with the climate zone. Moreover, the geographical regions in Burkina Faso  are divided into 13 parts, which making it difficult to analyse. We have so removed the reference to this variable from the Material and Methods section.

5.      Material and Methods, line 62-65: The sentence is rather difficult to understand and it is not clear which further (?) epidemiological factors are meant. Please clarify. In line 63-64 it should read “(HEV RNA and/or anti-HEV IgM positive)”.

We have followed the reviewer’s recommendation and modified the sentence as follow: “The patient’s sex and age, domicile (urban or rural), climate zone (arid, semi-arid or tropical savannah) as well as the year of sampling were analysed for their potential association with markers of an acute HEV infection (HEV RNA and/or anti-HEV IgM positive) and exposure to HEV (HEV RNA and/or anti-HEV IgM and/or anti-HEV IgG positive).”

6.      Results, line 80-81: The insertion “526 males (58.4%), mean age 15 years (±0.5, 95% CI: 14-16.1)” interrupts the reading flow. It would be better to characterise the analysed cohort in a first sentence and highlight the results in the following sentence.

We have modified the sentences as follow: « The cohort consisted of 526 males (58.4%), the mean age was 15 years (±0.5, 95% CI: 14-16.1). Of the 900 patient samples tested, we found 23/900 (2.6%; 95% CI: 1.6% - 3.8%) who had markers of an acute HEV infection. »

7.      Results, Figure 1: It would be helpful to highlight the boxes including the 23 acute infections by colour or bold outlines.

We have acted on the advice and we have modified the flow chart in figure 1 to highlight the 23 cases of recent HEV infection.

8.      Results, line 88: “14/23” indicates that all 23 samples could have been genotyped. However, 4 of these samples lacked HEV RNA. Therefore, “14/19 HEV RNA positive samples” might be more appropriate.

We have modified the sentence as suggested.

9.      Results, Figure 2: Are the sequences submitted to GenBank? If not, please, do this.

As indicated in the Results section, the sequences have been submitted to GenBank under accession number MK412900-MK412913. We have modified the Figure 2 adding the GenBank numbers.

10.  Results, Figure 2: The software which was used for generating the phylogenetic tree should be indicated.

We used ClustalX software to generate the phylogenetic tree. It is now indicated in the Methods section.

11.  Results, Figure 2: Additional African HEV-2b sequences (to which the colleagues also refer to in the discussion section in line 131) should be added to the phylogenetic analyses to better resolve the phylogenetic relationship of this subtype. And, please, indicate the sample numbers or ID of the patients at the dots of the phylogenetic tree.

The Figure 2 has been modified in the revised manuscript: we have added the genbank number and the sample number ID as requested. We have performed a phylogenetic tree with the only full length reference sequence available. The other genotype 2 strains described in Africa available are only shorter partial sequences that partially overlap with our sequences. Thus for clarity reasons, we prefer to perform phylogenetic tree with full length genotype 2 reference sequences available.

12.  Results, Figure 3: The number of samples and the number of IgG-positive cases for each group should be indicated.

We have completed the Figure 3.

13.  Discussion, line 114: The statement “Our data indicate that HEV genotype 2 was implicated in 2.6% of the cases of acute febrile jaundice” is not correct. Of the 23 cases only 14 could be genotyped.

We agree with the reviewer and we modified the sentence in Discussion as follows : « Our data indicate that HEV genotype 2 was implicated in 2.6% of the cases of acute febrile jaundice identified in Burkina Faso during 2013-2016. HEV genotype 2b was detected in samples that could be genotyped and that 18.2% of this population had anti-HEV IgG. »

14.  Discussion, line 125-127: As the samples were not tested for HBV and HCV as well as other possible pathogens the febrile icterus may not necessarily be attributable to HEV although markers of acute HEV were identified. This should be stated as a limitation of the study.

We have now stated in the revised manuscript that others possible pathogens, especially hepatitis B or C virus were not investigated.

15.  Discussion, line 129-133: The colleagues describe HEV-2 being identified in a number of African countries (line 130) and conclude that subtype 2b is established throughout the continent (line 133). It should be indicated more clearly, that the HEV-2 sequences to which they are referring are of subtype 2b which could be very nicely supported by including those sequences in the tree.

As explained previously, the other genotype 2 African sequences could not be included in the tree since they are shorter and partially overlap with our sequences. That’s why only the 2 full length genomes of HEV2 are include in the tree (The Mexican strain 2a and the Nigerian strain 2b). However, the phylogenetic analyses confirmed that they belong to genotype 2b.

16.  Discussion, line 137: The mean age of 14 is not equal to the one stated in line 80 (mean age=15).

 There was indeed an error in the Discussion section and we have modified the value of the mean age in the Discussion (mean age =15)

17.  Discussion, line 153-154: There is no hint to conclude that the source of the HEV-2b infections found in this study is different from previous cases in Burkina Faso, as no previous cases are reported.

The high seroprevalence found in butchers and the detection of HEV-3 in pigs in an earlier study indicates that probably asymptomatic or subclinical infections due to zoonotic transmission might be common in Burkina Faso.

However, the finding in this study indicates that acute, symptomatic HEV infections are rather due to faecal-oral transmission. Due to the lack of HEV surveillance in Burkina Faso the finding and interpretation of symptomatic cases is incomplete since no other study analysed HEV genotypes in Burkina Faso so far. Therefore, the conclusion should better highlight the confirmed circulation of subtype 2b in Burkina Faso.

We have now modified the sentence as follow:” We conclude that the genotype 2b infections identified in our patients indicate that the source of these sporadic cases of HEV is different from the HEV genotype found previously in pigs in Burkina Faso. If so, controlling this disease will also require better access to safe water and sanitation and improved personal hygiene.”

Reviewer 2 Report

AUTHORS

Manuscript ID: viruses-518672

Title: Hepatitis E virus infections among patients with acute febrile jaundice in Burkina Faso

This is an interesting study that presents data on the association between febrile icterus and hepatitis E in Burkina Faso. I advise publication contingent to minor revisions.

1.Authors mention a previous study [17], presenting evidence for HEV in Burkina Faso: “A recent study found HEV in both pigs and humans [17] but little is known about the epidemiology and the genotype causing hepatitis E in this country.” I would suggest to expand a bit on this study and explain what has been done and in what sense your study increases the knowledge in the region.

2.Authors goal was to “We have investigated the contribution of HEV to cases of acute febrile jaundice reported in the Burkina Faso Yellow Fever (YF) surveillance and identified the HEV genotype involved.” I disagree with the sentence as one cannot conclude on the contribution or role of HEV in the acute febrile jaundice, but only on the presence of HEV (or evidence of recent presence) in the sick. Authors have not excluded other hepatitis viruses or other type of hepatic disorders, to remove the confounders. Hence I would tone down the objectives.

3. Authors mention that “A recent study in the Democratic Republic of Congo found that 10.4% of 365 patients with febrile jaundice had markers of HEV infection [23], but this could be due to the increased frequency of HEV infection in a single town in August 2006 caused by an unnoticed hepatitis E outbreak”. What was the genotype involved in the outbreak in the Democratic Republic of Congo? Could the high case reporting be associated to a different (more pathogenic) genotype than in Burkina Faso? Please elaborate on this

4. Authors have shortly described the  distribution of genotype 2 in Africa and I think the manuscript would also gain in describing (shortly) the distribution of genotype 3 in Africa, since both the IgG and IgMs detected in this study could have came from HEV3.

5.Is there and explanation for a higher infection rate in young adults versus children? Professional exposure? Cultural reason?

6. I do agree with the conclusions however the manuscript should also consider the possibility of HEV3 contribution to the antibodies found.

Author Response

1.      Authors mention a previous study [17], presenting evidence for HEV in Burkina Faso: “A recent study found HEV in both pigs and humans [17] but little is known about the epidemiology and the genotype causing hepatitis E in this country.” I would suggest to expand a bit on this study and explain what has been done and in what sense your study increases the knowledge in the region.

We have now modified this part of the introduction as follow: A first survey of zoonotic risk of Hepatitis E virus (HEV) transmission in Ouagadougou, Burkina Faso was conducted. They found that anti-HEV antibodies were more prevalent in pork butchers than in the general population. Among slaughter-aged swine, HEV seroprevalence was of 80%, and HEV RNA was detected in 1% of pork livers. Phylogenetic analysis pointed out HEV genotype 3 [17] but little is known about the epidemiology and the genotype causing symptomatic hepatitis E in human in this country.

2.      Authors goal was to “We have investigated the contribution of HEV to cases of acute febrile jaundice reported in the Burkina Faso Yellow Fever (YF) surveillance and identified the HEV genotype involved.” I disagree with the sentence as one cannot conclude on the contribution or role of HEV in the acute febrile jaundice, but only on the presence of HEV (or evidence of recent presence) in the sick. Authors have not excluded other hepatitis viruses or other type of hepatic disorders, to remove the confounders. Hence I would tone down the objectives.

We have adopted a more nuanced objective and have modified the sentence as follows : « We have investigated the presence of HEV in the cases of acute febrile jaundice reported in the Burkina Faso Yellow Fever (YF) surveillance and identified the HEV genotype involved. »

3.      Authors mention that “A recent study in the Democratic Republic of Congo found that 10.4% of 365 patients with febrile jaundice had markers of HEV infection [23], but this could be due to the increased frequency of HEV infection in a single town in August 2006 caused by an unnoticed hepatitis E outbreak”. What was the genotype involved in the outbreak in the Democratic Republic of Congo? Could the high case reporting be associated to a different (more pathogenic) genotype than in Burkina Faso? Please elaborate on this

Unfortunately, the HEV genotype was not investigated in this previous study in Democratic Republic of Congo. Only the discussion stated that the majority of HEV cases occurred in a single town in August 2006. Thus, we cannot speculate that it was linked to a different (more pathogenic) genotype than in Burkina Faso.

4.      Authors have shortly described the  distribution of genotype 2 in Africa and I think the manuscript would also gain in describing (shortly) the distribution of genotype 3 in Africa, since both the IgG and IgMs detected in this study could have came from HEV3.

We have now added a short description on the HEV3 in Africa: “But, the zoonotic HEV genotype 3 was also observed in South Africa [32] Nigeria [27] Egypte [13] and Madagascar [33] suggesting that both waterborne and zoonotic transmissions occur in Africa.”

5.      Is there and explanation for a higher infection rate in young adults versus children? Professional exposure? Cultural reason?

In the present study, we found that the age was associated with an acute infection and also with IgG seroprevalence. All previous seroprevalence studies worldwide indicated that seroprevalence increase with age in relation with a higher frequency of exposure to the virus during life.  Similarly, a higher number of symptomatic infections in young adults versus children has always been reported for genotype 1 and 2 in developing countries [13]. The mechanism explaining this difference of symptomatic cases according to age and HEV genotype is not known and deserves further studies.  This point has been added lines 167-170.

6.       I do agree with the conclusions however the manuscript should also consider the possibility of HEV3 contribution to the antibodies found.

We have now added a sentence in the discussion section as follow:This prevalence of anti-HEV IgG could be linked to zoonotic transmission with genotype 3 and waterborne transmission due to genotype 2. “ (lines 176-178)